# Peer review of "The Mental Wellbeing of Child and Adolescent Mental Health Service (CAMHS) Workers in England: A Cross-Sectional Descriptive Study Reporting Levels of Burnout, Wellbeing and Job Satisfaction"

_healthcare, 2024, doi:10.3390/healthcare12040430_

Round 1

Reviewer 1 Report

Comments and Suggestions for Authors

Dear Authors 

I read the manuscript with interest.  Staff members' well-being, job satisfaction, and burnout in the Child and Adolescent Mental Health Service are important.  The manuscript is well-written.

The findings indicated that about a third of study respondents reported moderate (23%) or high (9%) work-related burnout; 39% reported moderate (32%) or high (7%) personal-related burnout. For client-related burnout, the proportion of responses classified as moderate (11%) or high (2%) was substantially lower. Most (72%) respondents reported moderate levels of well-being, 26% reported low levels of well-being, and only 2% reported high levels of well-being. About 70% of respondents endorsed that they either agreed or strongly agreed with statements such as “I feel fairly satisfied with my present job” and “I find real enjoyment in my work”.  

Job satisfaction was negatively related to work, personal and client burnout (all p-s<.005) and positively related to well-being. Wellbeing levels were negatively related to all three domains of burnout (all p-s <.005).

These results are very interesting and answer the first study objective, i.e. assess levels of staff burnout, well-being, and job satisfaction experienced in one UK CAMH service and how these are interrelated.

However, the study sample size is relatively small, which may affect the study's internal validity.  Also, several factors, such as age, working full-time versus part-time, and clinical versus non-clinical, were not significantly different with the CBI.  This weakened the study's findings and conclusions.  Therefore, the second study objective i.e. explore whether they differ by demographic and work-related factors (i.e., sex, age, clinical vs non-clinical professional role, and working pattern).  Thus, these results should be taken with caution. 

Author Response

Thank you for your careful consideration of our paper, we are glad that you enjoyed reading it. We do agree that the sample size being small is a limitation and we hope you agree that how we have addressed this as such in the 'Strengths and Limitations' section is adequate. We have also made this point clearer in the 'Conclusion' section. 

Reviewer 2 Report

Comments and Suggestions for Authors

1. You mention at different points in your paper that you will not be covering 'nature-based approaches'. I don't think you need to refer to this at all in the paper. It just leaves the reader wondering why not. You could put it in your conclusion as the next avenue for further reporting and leave it at that?

2. In the method or results could you tell the reader what the 'cut-off' points are for burnout and wellbeing and justify why you have set them there.

3. In table 1 would be helpful to include minimum and maximum scale scores.

4. Figures 2 needs reconsidering. Why not keep the correlations in a table. I'm not sure what the histograms add to your argument? Are they needed?

5. It might be me, but you say you are going to ANOVAs - do you? did you? Would regression be worth considering?

6. In the discussion you talk about the correlation between job satisfaction and burnout. Not surprisingly (as you point out) they are linked. What you cannot draw from correlations is any causation. This is where you may find regression helpful?

7. In the discussion you talk about comparison to other samples. This is helpful, but how do you know about these relationships? Did you  undertake t-tests?

Overall the paper read well. I think the research is important and has value. With some changes I think it should be published. I appreciated the opportunity of reading your research. 

Reviewer 3 Report

Comments and Suggestions for Authors

I read and analysed the brief report The Mental Wellbeing of Child and Adolescent Mental Health Service (CAMHS) Workers in England: Reported Levels of Burnout, Wellbeing and Job Satisfaction in detail.

The topic is state-of-the-art because the well-being of the staff in mental health services is important to us, especially those who work with children and adolescents.

The manuscript is clear, relevant for the field and presented in a well-structured manner. Also, the cited references are mostly recent publications (within the last 5 years) and relevant.

However, the authors stated in the Strengths and Limitations section that the strength of the study was the use of validated instruments. Therefore, it is necessary to show the reliability measures of all used instruments in the methodology and a statement whether the used instruments are free for use, or special permission was obtained from the author.

Author Response

Thank you for your careful consideration of our paper. We have added detail about the reliability of the measures used and indicated if they are free to use. 

Reviewer 4 Report

Comments and Suggestions for Authors

The Mental Wellbeing of Child and Adolescent Mental Health Service (CAMHS) workers’ in England: reported levels of burnout, wellbeing and job satisfaction

Thank you very much for allowing me to review the manuscript. I would like to provide you with a series of recommendations for your consideration regarding its publication:

Title: The title adequately captures the content of the manuscript. However, I would include the study type in it.

Abstract: The abstract should encompass the study's methodology, as well as inclusion and exclusion criteria.

Methodology:

- The section "the current study" should be included in the methodology section. Specify the study design. It is not clear to me what type of design the researchers have employed.

- Include the calculation of the sample size and response rate.

- Provide a brief description of the psychometric properties of the assessment instruments used.

- Place the objectives of the work at the end of the introduction, not in the methodology section.

- The "data analysis" section does not provide sufficient information to replicate and understand the study adequately.

Results:

- Include a table describing the sociodemographic characteristics of the sample.

- Overall, it is not clear what this study contributes in the results. Please clarify and provide analyses that offer more significant and relevant contributions to scientific knowledge.

Discussion:

- In line 245, bibliographic references 23 and 24 should be in the same bracket.

- I believe the results have not been adequately discussed in the discussion section.

- Include more bibliographic references.

- Add a section indicating the implications for clinical practice.

Author Response

Thank you for your careful reading of our paper. We have responded to your comments in the attached document. 

Round 2

Reviewer 4 Report

Comments and Suggestions for Authors

Thank you very much for allowing me to review the manuscript again.

The authors have addressed my comments.